# Differences in Cognitive and Non-Cognitive Results between Only-Child and Non-Only-Child Children: Analysis of Propensity Scores Based on Large-Scale Assessment

**DOI:** 10.3390/children9060807

**Published:** 2022-05-30

**Authors:** Dexuan Zhao, Zhuang He, Yi Tian, Hongyun Liu

**Affiliations:** 1Faculty of Psychology, Beijing Normal University, Beijing 100875, China; zhaodexuan@mail.bnu.edu.cn; 2School of Educational Sciences, Guiyang University, Guiyang 550005, China; hezhuang@mail.bnu.edu.cn; 3Beijing Academy of Educational Sciences, Beijing 100045, China; tianyi@mail.bnu.edu.cn

**Keywords:** only child, cognitive and non-cognitive outcomes, propensity-score matching

## Abstract

Based on the data of 3561 fifth-grade and 4062 eighth-grade students from the Beijing Assessment of Educational Quality in China, the present study used a propensity-value matching model to scientifically analyze only-child and non-only-child children in primary and secondary schools. Female differences in cognitive outcomes (linguistic performance) and non-cognitive outcomes (teacher-student relationships, peer relationships, and emotional management) were also evaluated. The results of the study were as follows. First, fifth-grade only-child students had a higher linguistic performance compared to that of their non-only-child counterparts, and the same result was found for eighth-grade students. Second, fifth- and eighth-grade only-child students had good teacher-student relationships that were not significantly different from those of their non-only-child counterparts. Third-, fifth-, and eighth-grade only-child students had significantly better peer relationships and emotional management compared to these parameters in their non-only-child counterparts.

## 1. Introduction

Studies of only-child students have long attracted attention from all aspects of education, at home and abroad. The earliest foreign studies initially regarded only-child students as “problem children” or “special children”. The earliest scholar to openly raise the potential issue of the only child was the well-known American psychologist, Granville Hall, who posited in 1896 that “the only child is a disease in itself”. In 1898, American psychologist, Bo Hannong, used a questionnaire to conduct a case study of an only child and analyzed the specificity of the only child. The study concluded that there are two types of extreme personality characteristics in the only child, claiming that such children are significantly inferior to ordinary children in terms of physical, intellectual, and social abilities. Since 1928, Fenton, Hook, Wuster, Gilford, and others in the United States have published a series of research reports on potential issues in only-child individuals. In many aspects, such as in personality characteristics, only-child individuals are not much different from non-only-child individuals. After the 1950s, researchers’ attitudes and views on the only child changed. More and more researchers have begun to emphasize the potential advantages of only-child individuals. Especially in the 1960s through the 1980s, a more unified view of the only child occupied Western studies, suggesting that only-child individuals are superior to non-only-child individuals in terms of intelligence [1], although the performances are not the same under different conditions [2].

Falbo et al. reviewed research on the only child as early as 1977 and later updated this research in terms of academic achievements, personality characteristics, and social behaviors of the only child and non-only child in 2012. Falbo and Polit [3] conducted a meta-analysis of 115 studies—published between 1925 and 1984—on the only child, and found that the only child was superior to the non-only child in terms of personality characteristics such as control, autonomy, and psychological maturity, but not any social aspects. Polit [4] conducted a review of studies on the only child and personality development and found that the only child was significantly better than other groups in terms of achievement motivation and personal adaptation. Poston [5] compared academic achievements and personality characteristics and found that the only child performed significantly better academically than the non-only child, but scored similarly in terms of personality characteristics. Meredith [6] compared the self-concept and social consequences of the only child and found that there were no significant differences compared to these attributes in non-only-child individuals. Some researchers have compared the academic performances, personalities, and physical conditions of only-child and non-only-child students in grades 3–6 in China and found that only-child students had better academic performances than those of non-only child students, whereas there was little difference in terms of personality or physical characteristics. Chuanwen Wan [7] studied differences in personality characteristics between the only child and non-only child (grades 1, 3, and 5) and found that learning motivation was greater in the non-only child, whereas there were no differences in terms of interpersonal skills or attitudes. Liu et al. [8] studied the academic performances of only-child and non-only-child students and predicted that only-child students would be more likely to go to college than non-only children. Falbo and Hooper [9] used SCL-90 to comprehensively analyze the mental health of only-child children and found that the mental health of only-child students was better than that of non-only-child students.

It has been more than 30 years since China began to implement the one-child policy in 1979. There have also been many domestic studies on differences between only-child and non-only-child individuals since that time. However, when it comes to cognitive performance, the conclusions of these studies are inconsistent. Li Feng and Xin Tao [10] compared the math scores of only-child and non-only-child students in junior high school using propensity-value matching and found that before matching, only-child students had better mathematical results compared to those of non-only-child students, whereas there was no significant difference between the two after matching. Huang Lin and Wen Dongmao [11] compared the study habits and living conditions of only-child and non-only-child university students and found that the economic and cultural backgrounds of the families of only-child students were significantly better than those of non-only-child families, and that only-child students enjoyed a higher quality of higher education as well; education costs were higher and lack of learning initiative led to reduced academic performance in non-only-child students.

Regarding non-cognitive performance, the conclusions of these studies in China are also inconsistent. A survey by Cui Yuzhong [12] showed that there was no significant difference in the teacher-student relationship between only-child and non-only-child students in primary schools, which was a finding that was recapitulated for students in junior high school in a survey by Zhao Qing [13]. Liu Haiying [14] surveyed the peer relationships of only-child students in primary and secondary schools and found that the overall status of peer relationships between only-child students in primary school versus those in middle school were similar, such that they were both welcome and unpopular. However, there is a significant gender difference in peer interactions among only-child students in primary and secondary schools, such that peer relationships between girls are significantly better than those between boys. At the middle school level, the number of unwelcome rural only-child children was significantly higher than that of urban only-child children. In addition, some studies have shown that there is no significant difference in emotional adaptation between only-child and non-only-child individuals, and that, in some aspects, only-child children have advantages over non-only-child individuals [15].

Taken together, many studies have been conducted on the academic performance, personality traits, emotions, motivations, and physical and mental conditions of only-child individuals at home and abroad. These studies can be roughly divided into two perspectives. One perspective considers the only child as the research object and explores personality characteristics, socialization, and social adaptations. The other perspective studies only-child variables to explore the impact of the only-child phenomenon on families and society [16]. The present study focused on the first perspective to investigate differences between only-child children and non-only children in terms of cognitive results (academic performances) and non-cognitive results (teacher-student relationships, peer relationships, and emotional management).

However, although past comparisons of cognitive and/or non-cognitive results of only-child individuals have rarely systematically considered the underlying factors of such results, some recent studies have begun to investigate such factors. Liu et al. [17] studied the impact of the demographic characteristics of only-child children on cognitive and non-cognitive outcomes. The differences in cognitive and non-cognitive outcomes for only-child children can be rooted in gender, geographical region, parental education, parental expectations, family socioeconomic status, and family structure. Other studies have controlled for potential socio-demographic variables (such as gender, age, learning level, economic status, family structure, and mobility status) to compare differences in psychological behaviors (such as mental health and academic achievement satisfaction) between only-child and non-only-child individuals. There are distinct differences in psychological and behavioral characteristics between only-child and non-only-child individuals [18]. Therefore, simply comparing the means of parameters without performing statistical tests (e.g., via t-tests, analyses of variance) on various background factors between only-child and non-only-child individuals may produce biases; such research methods ignore the “sample non-random selection problem” that must be solved when causal inferences are made in social science research. In order to test whether there are statistically significant differences in cognitive and non-cognitive results between only-child and non-only-child individuals, and whether these differences are due to the only-child parameter, we used propensity-score matching in the present study. This is an alternative strategy for random allocation. When random-allocation experiments are not feasible, propensity-score matching can minimize the impact of confounding variables on the results. Therefore, propensity-score matching is often used to identify the effects of experimental processing in research areas where social processing cannot be randomly assigned.

## 2. Materials and Methods

### 2.1. Research Subjects and Sampling

The research subjects included a sample population of 3561 fifth-grade students (Appendix A) and a sample population of 4062 eighth-grade students (Appendix A). Sampling was mainly based on a combination of multi-stage random sampling and stratified cluster sampling. The specific sample distribution is shown in Table 1.

### 2.2. Research Tools

#### 2.2.1. Subject Test Papers

Subject test papers included fifth- and eighth-grade quizzes. The academic test papers were made by subject-proposition expert teams following the compulsory educative language-curriculum standards in strict accordance with the formulating plans, blueprints, propositions, review questions, pre-tests and analyses, test papers, and score-line developments. The language discipline included areas such as literacy, reading, and writing. Table 2 indicates that the academic test papers all had positive educational-measurement indicators.

Regarding content validity, subject-review experts were hired according to curriculum standards and the specific requirements of the “Content Validity Table for Academic Level Tests,” from the guiding ideology and basis of the subject test papers, the structure of the test papers, the overall evaluation of the test questions, the guidance of the test papers, and the actual results. Examination of subject test papers was evaluated from five aspects (e.g., sexuality), using a five-point Likert scoring method.

#### 2.2.2. Teacher-Student Relationship Questionnaire

The questionnaire that we used was initially prepared by Pianta [19] and was revised by Wang Yun [20]. This questionnaire contains 28 items, divided into three dimensions: intimacy, conflict, and responsiveness. Teacher reports were also used. The five-point scale assessed the degree of compliance of the situation described, and scores ranged from 1–5 points from “completely non-compliant” to “fully qualified.” Qu Zhiyong [21] revised this questionnaire from the perspective of student reports. The revised questionnaire consists of 23 items, which are divided into four dimensions of intimacy, conflict, support, and satisfaction. Our present study abridged this revised questionnaire, and ultimately retained 16 items with four dimensions: atmospheric intimacy, conflict, support, and satisfaction.

The reliability of internal consistency of each questionnaire was 0.76, 0.72, 0.75, and 0.72, indicating that there was acceptable internal consistency. The use of confirmatory factor analysis showed that this model had a favorable index of goodness of fit, and that all non-standardized factor loads reached a significant level and maintained stability between different gender samples and samples from different regions. Overall, this analysis demonstrated that this model had acceptable structural validity.

#### 2.2.3. Peer Relationship Questionnaire

The peer relationship scale used in this study was revised from the Peer Relationship Scale for Children and Adolescents compiled by Professor Guo Boliang of the Chinese University of Hong Kong. There were 22 items in the original questionnaire, and scores reflected the following: 1, not so; 2, sometimes; 3, often; and 4, always like this. The higher the score, the worse the peer relationship (after reverse scoring). The internal consistency coefficient of the questionnaire was 0.71, which indicated positive reliability. Considering the length of the questionnaire and the time limit for answering the questions, the project team deleted the original questionnaire after the pre-test. According to the results of the confirmatory factor analysis, the questions with a lower load were deleted and 15 questions were ultimately retained, of which 6 of the questions belonged to the peer-anxiety dimension and 9 of the questions belonged to the peer-acceptance dimension.

The reliability of internal consistency of each questionnaire was 0.575 and 0.867, indicating that they had acceptable internal consistency. The use of confirmatory factor analysis showed that this model had a favorable index of goodness of fit, and that all non-standardized factor loads reached a significant level and maintained stability between samples of different genders and samples from different regions. Overall, this analysis demonstrated that this model had positive structural validity.

#### 2.2.4. Emotion Management Questionnaire

The Emotion Management Scale used in this study was revised from the Chinese version of the Baron Emotional Intelligence Scale (Adolescent Edition). The original scale had a total of 60 topics, divided into six dimensions, which were internal, interpersonal, adaptive, stress management, overall emotional intelligence, and general mood. Considering the purpose of the survey and the time required for answering each question, the content of the first four dimensions was specifically examined. After analyzing the pre-test data, it was finally decided to reduce the questionnaire to 18 questions, according to the following scoring system: 1, rare; 2, rare; 3, frequent; and 4, usually. Each item was evaluated, and the score of each item ranged from 1–4 points. Higher scores were indicative of better emotional management (after reverse scoring).

The reliability of internal consistency of each questionnaire was 0.88, 0.70, 0.62, and 0.87, indicating that there was acceptable internal consistency. The use of confirmatory factor analysis showed that this model has a favorable index of goodness of fit, and that all non-standardized factor loads reached a significant level and maintained stability between samples of different genders and samples from different regions. Overall, this analysis demonstrated that this model had positive structural validity.

### 2.3. Variable Selection

The dependent variable consisted of whether the individual was an only child or a non-only child (1 for only child, 0 for non-only child), Covariates included gender, household registration, children moving, region, suburban area, scale, and socioeconomic status; the first six of these covariates were virtually coded, whereas socioeconomic status was a continuous variable. Processing variables included academic performance, teacher-student relationship, peer relationship, and emotional management. Among them, the academic results were scaled by IRT, which is a standard score of 0–800; the other three variables were scored by IRT to form a standard nine-point score (i.e., ranging from 1–9).

### 2.4. Statistical Analysis

The psmatch2 package of Stata14.0 software (StataCorp LLC, 4905 Lakeway Drive, College Station, TX, USA) [22] was used to calculate the propensity values. The analysis procedure followed the review method of Su Yusong [23]. The corresponding descriptive statistics and t-test analyses were performed using SPSS20.0 software.

## 3. Results

### 3.1. Differences before Matching

Before matching, there were significant differences between fifth and eighth graders in Chinese language scores, literacy and writing, reading and accumulation, and assignments between only-child and non-only-child students. Details are shown in Table 3.

Before matching, there were significant differences between these groups in the overall levels of teacher-student relationships with intimacy, conflict, and supportive dimensions. There were no significant differences in terms of satisfaction dimensions. Details are shown in Table 4.

Before matching, there were significant differences between these groups in emotional management dimensions. Details are shown in Table 5.

Before matching, there were significant differences between these groups in terms of peer relationships dimensions. Details are shown in Table 6.

### 3.2. Estimated Tendency Value

Estimating causal effects using propensity-value matching includes estimating propensity values and matching analysis. In general, regression methods such as logit and probit models can be used to estimate propensity values, and balance coincidence tests and sensitivity analysis should also be performed.

#### 3.2.1. Logit Regression

For the calculation of propensity scores, logit and probit models were used for regression analysis to calculate how only-child and non-only-child students were affected by variables such as gender, household registration, region, scale, urban vs. rural, and the socioeconomic status of the family.

Table 7 shows the estimated results of the propensity score models of the logit model and the probit model. The conclusions drawn by the two groups of models were relatively consistent, and the correlation between the propensity scores of the two groups was 0.94. The results have favorable stability.

After analyzing the results shown in Table 7, it was found that from the perspective of statistical inference, the probability of a male student being an only child was greater than that of female students. The probability of urban students being an only child was greater than that of students from other categories of households. The probability that a student from a large-scale school was an only child was greater than that of students from other school sizes. A non-family student was more likely to be an only child than a relocated student. However, the influence of suburban counties on grouping variables did not reach statistical significance.

Overall, the regression model results also verified the non-negligible heterogeneity problem between only-child and non-only-child students, demonstrating the necessity of using propensity-score pairing in the present study.

#### 3.2.2. Estimating Treatment Effects

The nearest-neighbor matching method, the radius matching method, the kernel matching method, and the Mahalanobis distance-matching method were used to estimate the causal effects of the only-child factor on the Chinese performance of the included students. In addition, in order to ensure the stability of the estimation results, after each method was used for estimation, we used a Bootstrap repeated-sampling method to perform stability tests and found that the stability of the results was very favorable (Table 4). After considering the two issues of heterogeneity and sample selectivity bias, the only-child factor had a positive impact on the academic development of the included students, whereas this factor only had an average effect on the literary and academic achievements of the included students. The effect (ATT) was about 35; if the only-child students in the elementary school were non-only children instead, the average effect (ATU) of their literary achievements was about 28; the only-child factor had an overall effect on the academic achievements of all students and this effect was around 30. The details of these data and analyses are shown in Table 8.

The results obtained by the four matching methods were similar to one another, indicating that propensity-score matching was robust. The following only presents the results of the nearest-neighbor matching method.

#### 3.2.3. Balance-Coincidence Inspection

For the balance test, it was necessary to test the difference between the means of the covariates before and after matching. From Table 9, it can be seen that except for gender, suburbs, and scale, most of the covariates were matched and any biases were reduced by more than 85%. Based on the results of the balance test, it can be inferred that the choice of matching variables and matching methods in this study were appropriate. After matching, only-child and non-only-child students had a high degree of consistency in terms of household registration, region, size, and family socioeconomic background.

Table 9 above summarizes the estimates of key covariates. According to the results, there were significant differences in gender, urban vs. rural, family type, and socioeconomic status of family income between only-child and non-only-child students before matching. These factors confounded the differences in emotional adaptation between only-child and non-only-child students. After the nearest-neighbor matching of propensity scores, the differences between only-child and non-only-child students in most of the above variables were not statistically significant. Before matching, the only-child group had a significantly higher propensity score than that of the non-only-child group; after matching, there was no difference between only-child and non-only-child students. The above results indicate that nearest-neighbor matching was relatively successful. Additionally, it can be seen from Figure 1 and Figure 2 that the distribution of the tendency scores of the post-matching processing group and the control group was more consistent.

For the coincidence test, the psgraph program was used to check the overlap of the propensity values of the experimental group and the control group. The above figure shows that the samples of the experimental group had corresponding control samples in the range of 0–1 and that the overlap was acceptable.

Figure 3 presents a bar chart of the propensity scores of only-child and non-only-child students calculated by the logit model. We found that the distribution of propensity scores of the two groups had a favorable overlap; for samples with propensity scores above 0.5, the number of samples of the only-child group was greater than that of the non-only-child group. In the 0.5 sample, the number of samples in the only-child group was smaller than that of the non-only-child group.

#### 3.2.4. Sensitivity Analysis

The results of Wilcoxon’s symbolic rank test (sig+ and sig-; two columns) and the Hodges–Lehmann point-estimation test (t-hat+, t-hat-, CI+, and CI-; four columns) show that contact processing of the two matching units occurred when the difference in the ratio was only 1.3-fold, and that the original conclusions about the treatment effect were changed. Details are shown in Table 10.

### 3.3. Difference Test after Matching

After matching, the only-child and non-only-child groups had significant differences in total language scores, literacy and writing, reading and accumulation, and assignments. Details are shown in Table 11.

After matching, there was no significant difference in the overall levels of teachers-student relationships or intimacy, conflict, or support dimensions. There were significant differences in the satisfaction dimension. Details are shown in Table 12.

After matching, there were significant differences in the emotional management dimensions. Details are shown in Table 13.

After matching, there were significant differences in the peer relationship dimensions. Details are shown in Table 14.

## 4. Discussion

In the present study, in terms of the differences in cognitive results between only-child and non-only-child students, fifth-grade only-child students had a higher language score than that of their non-only child counterparts, and this finding was recapitulated in eighth-grade students. Additionally, the total Chinese language scores in fifth- and eighth-grade students and the scores in each content area were at relatively high levels. For example, the fifth-grade only-child group’s mean math score was 540.5 and it was above 500 in all fields. At the same time, before and after matching, the academic performance of the only-child group was significantly higher than that of the non-only-child group, which is consistent with previous research conclusions [24]. The reasons that only-child students are outstanding in school are also due to advantages in terms of family, school, and social resources [17].

In terms of the differences in non-cognitive results, fifth- and eighth-grade only-child students had significantly better teacher-student relationships compared to those of their non-only-child counterparts. For example, the fifth-grade only-child group’s teacher-student relationship score was 6.21 and that of eighth graders was 5.03, both of which were at a high position of nine. Relatively speaking, the fifth-grade only-child group’s teacher-student relationship was better than that of the eighth-grade only-child group. However, there was no significant difference in the teacher-student relationship between only-child and non-only-child students after matching, which is mostly consistent with the conclusions of domestic researchers [12]. However, further analysis revealed that the satisfaction degree of fifth-grade only-child students was significantly better than that of non-only-child students; the eighth-grade only-child group was significantly more intimate than that of the eighth-grade non-only-child group. This may be because, compared to primary school classmates, only-child children get more attention [25], and compared to middle school classmates, only-child children are more confident and cooperative [26]. However, the underlying reasons for these results need to be further studied.

In the fifth and eighth grades, only-child students had significantly better peer relationships and emotional management than those of their non-only-child counterparts. For example, the peer-to-peer relationship score of the fifth-grade only-child group was 5.29, which was a high score of nine; the eighth-grade only-child group’s overall emotional management score was 5.41, which was a high score of nine. At the same time, peer relationships and emotional management of the only-child group were significantly better than those of the non-only-child group. This is inconsistent with previous conclusions. Falbo’s meta-analysis of 30 articles on sociality and regulation showed that there was no significant difference between peers and personal emotional management between only-child and non-only-child individuals [27]. Studies have shown that only-child children have better interpersonal affinities than those of non-only-child children [28]. Whether this is one of the reasons that only-child children have better peer relationships and emotional management than non-only-child children requires further study. In addition, our present study focused on differences in cognitive function and personality characteristics between only-child and non-only-child students. However, some researchers have studied and explored the mechanisms behind these differences between only-child and non-only-child individuals and have proposed that the family environment may affect differences in neurological and brain mechanisms between these two groups [29].

Due to social and moral reasons, it is impossible to randomly assign subjects to the only-child group or non-only child group based on the experimental control method; it is necessary to eliminate the interference of covariates such as gender, urban and rural areas, and socio-economic status. Therefore, the research adopted the method of propensity-value matching to control the above variables to a great extent. Compared to previous studies, the research is based on the research design of quasi experiments, and the propensity-value matching model was established to effectively control the problem of sample selection deviation and better control the estimation deviation caused by sample heterogeneity. As a result, the precision of the research was greatly improved more likely ensuring the research results to be true and effective.

However, the reliability of propensity value largely depends on the observed covariates, which is limited by the number of covariates in our database. Through sensitivity analysis, it can be seen that other variables may also affect the processing effect. Therefore, in future research, confounding covariates should be investigated as much as possible to make the propensity-value matching model more robust. Without the control of confounding covariates, the observed results of only-child and non-only-child groups in academic performance, teacher-student relationships, peer relationships, and emotional management may be spurious. From the logit regression of propensity scores, we can see that the only-child group is largely affected by many factors, such as gender, household registration, urban vs. rural, and socioeconomic status. For example, we found that boys were more likely to be an only child than girls, and students living in urban areas were more likely to be an only child than students in other household categories. Students in urban schools were more likely to be to be an only child than students in rural schools. Students with a higher socioeconomic status also had a greater likelihood to be an only child than students with a lower socioeconomic status. Using propensity-value matching, we were able to control for the above variables to a large extent. Compared to previous studies, our research design based on quasi-experiments was used to establish a propensity-value pairing model, which effectively controlled for sample-selection bias. The estimation bias caused by the heterogeneity of samples was also greatly mitigated and the precision of our research was improved. Therefore, our present results are more likely to be statistically valid [30]. However, the reliability of the propensity value depends to a large extent on the observed covariates, which were limited by the number of covariates in our database. Through sensitivity analysis, it can be seen that other variables may also have affected this processing effect. Therefore, we should investigate as many confounding covariates as possible in future research to make the propensity-matching model more robust.

By using scientific research methods, this study explored the differences between the cognitive results and non-cognitive results of the only child and non-only child children in primary and secondary schools. It elaborated on the current situation of the differences between the only child and non-only child and focused on the education and training of the only child based on the current situation, providing an empirical basis for the formulation of social education policies.

## Figures and Tables

**Figure 1 children-09-00807-f001:**
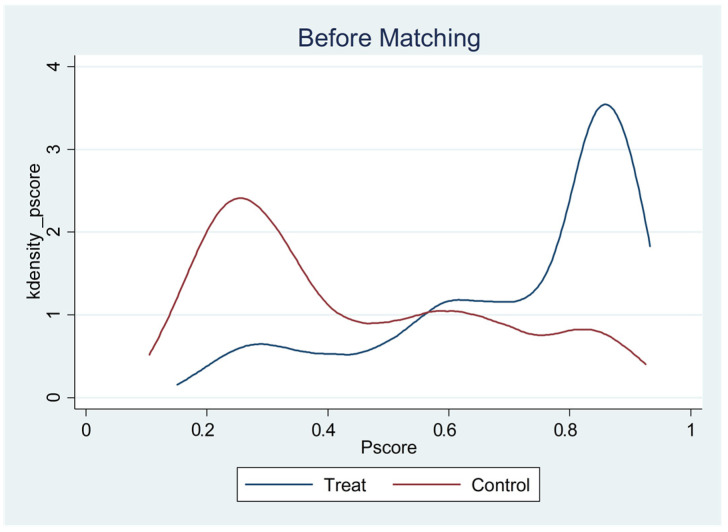
Distribution of propensity scores for the pre-matching treatment and control groups.

**Figure 2 children-09-00807-f002:**
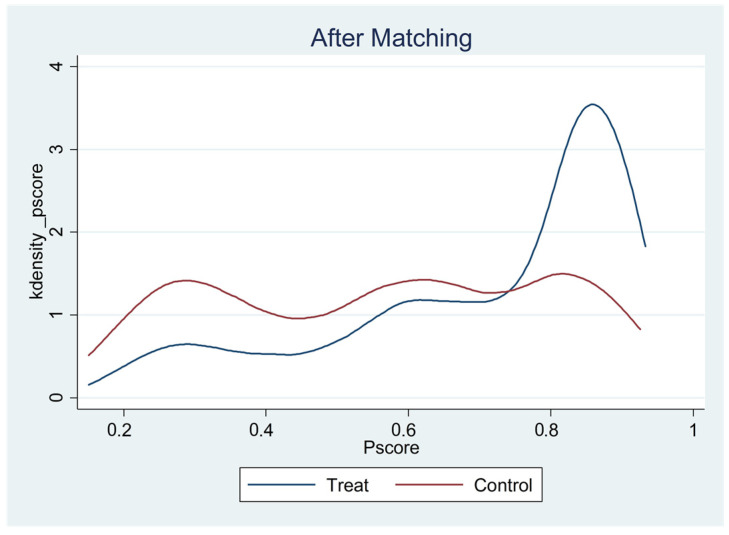
Distribution of propensity scores for the post-matching treatment and control groups.

**Figure 3 children-09-00807-f003:**
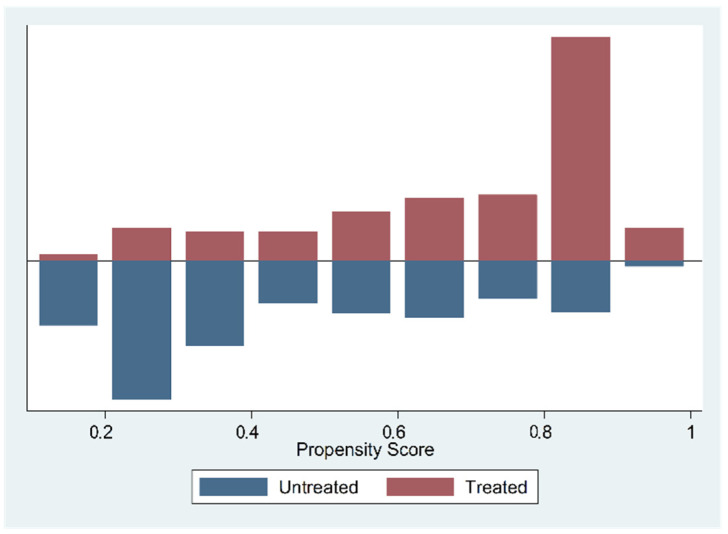
The coincidence of the sample propensity values of the post-matching treatment and control groups.

**Table 1 children-09-00807-t001:** Distribution of only-child and non-only-child students in fifth and eighth grade.

Grade	Non-Only Child		Only-Child	
	People	Percentage	People	Percentage
Fifth grade	1495	42.0%	2066	58.0%
Eighth grade	1460	35.9%	2602	64.1%

**Table 2 children-09-00807-t002:** Reliability and validity of academic test papers.

	Reliability		Validity	
	Internal-Consistency Coefficient	Half-Reliability	Content Validity	Total Correlation Coefficient
Fifth grade Chinese paper	0.87	0.72	≥4.50	0.13–0.49
Eighth grade Chinese paper	0.82	0.72	≥4.50	0.18–0.61

**Table 3 children-09-00807-t003:** Academic performance dimensions before matching.

		Fifth Grade				Eighth Grade			
		Mean	SD	t	*p*	Mean	SD	t	*p*
Total score	Only child	528.4	129.0			488.2	119.8		
	Non-only child	486.7	132.6	9.368	0.000	465.0	114.3	6.106	0.000
Literacy and writing	Only child	518.8	149.3			531.8	136.8		
	Non-only child	472.1	153.7	9.060	0.000	508.6	133.8	5.271	0.000
Reading and accumulation	Only child	525.9	129.2			500.1	113.3		
	Non-only child	484.7	132.1	9.275	0.000	480.7	108.7	5.351	0.000
Assignment	Only child	547.6	116.4			455.4	138.4		
	Non-only child	511.4	122.4	8.914	0.000	426.3	131.8	6.636	0.000

**Table 4 children-09-00807-t004:** Teacher-student relationships with intimacy, conflict, and supportive dimensions before matching.

		Fifth Grade				Eighth Grade			
		Mean	SD	t	*p*	Mean	SD	t	*p*
Intimacy	Only child	5.62	1.94			5.02	1.94		
	Non-only child	5.56	1.94	0.918	0.358	4.90	1.90	1.936	0.053
Conflicting	Only child	6.47	2.69			4.99	2.50		
	Non-only child	6.41	2.62	0.663	0.507	4.94	2.50	0.583	0.560
Supportive	Only child	6.36	2.20			5.00	1.95		
	Non-only child	6.25	2.16	1.472	0.141	4.90	1.91	1.573	0.116
Satisfaction	Only child	5.59	1.95			4.99	1.75		
	Non-only child	5.43	1.93	2.450	0.014	4.96	1.77	0.609	0.543
Teacher–student relationship	Only child	6.10	2.08			5.00	1.93		
	Non-only child	6.00	2.04	1.394	0.164	4.94	1.91	1.038	0.299

**Table 5 children-09-00807-t005:** Emotional management dimensions before matching.

		Fifth Grade				Eighth Grade			
		Mean	SD	t	*p*	Mean	SD	t	*p*
Within the individual	Only child	5.24	2.43			5.14	2.64		
	Non-only child	4.87	2.41	4.479	0.000	4.59	2.58	6.433	0.000
Interpersonal management	Only child	5.19	2.00			5.37	1.81		
	Non-only child	4.89	1.98	4.496	0.000	4.97	1.73	6.856	0.000
Adaptability	Only child	4.90	1.30			5.27	1.46		
	Non-only child	4.72	1.26	4.147	0.000	4.99	1.40	5.907	0.000
Stress management	Only child	5.24	2.22			5.38	2.03		
	Non-only child	4.91	2.20	4.404	0.000	4.89	1.97	7.662	0.000
General level of emotional management	Only child	5.19	1.99			5.37	1.91		
	Non-only child	4.87	1.96	4.799	0.000	4.93	1.84	7.241	0.000

**Table 6 children-09-00807-t006:** Peer relationships dimensions before matching.

		Fifth Grade				Eighth Grade			
		Mean	SD	t	*p*	Mean	SD	t	*p*
Peer anxiety	Only child	5.11	1.39			4.99	1.38		
	Non-only child	4.90	1.34	4.473	0.000	4.88	1.33	2.454	0.014
Peer acceptance	Only child	5.27	2.54			5.12	2.41		
	Non-only child	4.96	2.46	3.640	0.000	4.92	2.38	2.668	0.008
Peer relationship	Only child	5.15	2.04			5.07	1.99		
	Non-only child	4.88	1.96	3.916	0.000	4.91	1.95	2.427	0.015

**Table 7 children-09-00807-t007:** Table of logit and probit regression model results for covariates between only-child and non-only-child students (using fifth graders as an example).

Common Variable	Logit Model				Probit Model			
	Coef.	Std. Err.	z	*p* > z	Coef.	Std. Err.	z	*p* > z
XB1	0.23	0.09	2.59	0.01	0.14	0.05	2.60	0.01
HJ2	−0.74	0.14	−5.28	0.00	−0.43	0.08	−5.24	0.00
HJ3	−1.51	0.14	−10.87	0.00	−0.89	0.08	−10.92	0.00
HJ4	−2.28	0.19	−11.82	0.00	−1.37	0.11	−11.94	0.00
DY2	0.06	0.17	0.34	0.73	0.04	0.10	0.40	0.69
DY3	−0.31	0.15	−2.04	0.04	−0.19	0.09	−2.08	0.04
GM2	0.05	0.11	0.46	0.65	0.03	0.06	0.51	0.61
GM3	−0.16	0.16	−1.00	0.32	−0.10	0.10	−1.03	0.30
SQ1	−0.15	0.16	−0.93	0.35	−0.10	0.10	−1.00	0.32
CQ1	0.22	0.15	1.43	0.15	0.13	0.09	1.47	0.14
SES	0.30	0.05	5.83	0.00	0.18	0.03	5.84	0.00

**Table 8 children-09-00807-t008:** Estimated processing effect of different matching methods for fifth-grade samples.

Nearest-Neighbor Matching	Treated	Controls	Difference	S.E.
ATT	540.5	511.9	28.6	6.3
ATU	494.8	521.5	26.8	7.2
ATE	/	/	27.8	4.8
**Radius Matching**	**Treated**	**Controls**	**Difference**	**S.E.**
ATT	540.5	502.5	38.0	7.9
ATU	494.4	522.9	28.5	6.2
ATE	/	/	34.2	8.3
**Kernel Matching**	**Treated**	**Controls**	**Difference**	**S.E.**
ATT	540.5	501.5	39.0	8.6
ATU	494.8	522.7	27.9	5.3
ATE	/	/	34.5	8.5
**Mahalanobis Distance Matching**	**Treated**	**Controls**	**Difference**	**S.E.**
ATT	540.5	509.0	31.5	7.8
ATU	494.8	521.9	27.1	7.9
ATE	/	/	29.7	5.7

S.E.: standard error. ATT: average treatment effect on the treated. ATU: average treatment effect on the untreated. ATE: average treatment effect.

**Table 9 children-09-00807-t009:** Balance test of covariates in propensity-score matching (using fifth graders as an example).

		Mean		Standard Deviation Reduction		T-Test	
		Treated	Control	Standard Deviation (%)	Decrease (%)	t	*p* > t
XB1	U (Unmatched)	0.53	0.51	3.6	28.0	0.95	0.34
	M (Matched)	0.53	0.51	2.6		0.76	0.45
HJ2	U (Unmatched)	0.19	0.15	10.4	78.5	2.68	0.01
	M (Matched)	0.19	0.18	2.2		0.62	0.53
HJ3	U (Unmatched)	0.14	0.17	−8.3	53.0	−2.19	0.03
	M (Matched)	0.14	0.16	−3.9		−1.16	0.25
HJ4	U (Unmatched)	0.14	0.54	−94.0	99.7	−25.35	0.00
	M (Matched)	0.14	0.14	0.3		0.10	0.92
DY2	U (Unmatched)	0.34	0.25	19.1	81.5	4.94	0.00
	M (Matched)	0.34	0.36	−3.5		−0.98	0.33
DY3	U (Unmatched)	0.18	0.35	−38.0	95.7	−10.11	0.00
	M (Matched)	0.18	0.17	1.6		0.54	0.59
GM2	U (Unmatched)	0.36	0.45	−18.8	81.9	−4.91	0.00
	M (Matched)	0.36	0.34	3.4		1.01	0.31
GM3	U (Unmatched)	0.07	0.14	−24.6	84.9	−6.61	0.00
	M (Matched)	0.07	0.06	3.7		1.36	0.17
SQ1	U (Unmatched)	0.18	0.57	−89.0	98.0	−23.79	0.00
	M (Matched)	0.18	0.18	−1.8		−0.59	0.56
CQ1	U (Unmatched)	0.46	0.43	5.6	39.8	1.45	0.15
	M (Matched)	0.46	0.47	−3.4		−0.97	0.33
SES	U (Unmatched)	0.20	−0.39	61.2	95.0	16.03	0.00
	M (Matched)	0.20	0.23	−3.1		−0.90	0.37

**Table 10 children-09-00807-t010:** Sensitivity-analysis table for propensity-score matching (using fifth graders as an example).

Gamma	sig+	sig-	t-hat+	t-hat-	CI+	CI-
1	0.00	0.00	25.60	25.60	17.21	34.01
1.1	0.00	0.00	18.37	32.85	9.99	41.31
1.2	0.00	0.00	11.76	39.47	3.38	47.97
1.3	0.09	0.00	5.77	45.55	−2.67	54.05
1.4	0.48	0.00	0.18	51.18	−8.27	59.73
1.5	0.88	0.00	−4.93	56.40	−13.53	64.99
1.6	0.99	0.00	−9.80	61.27	−18.34	69.95
1.7	1.00	0.00	−14.33	65.84	−22.91	74.57
1.8	1.00	0.00	−18.54	70.15	−27.18	78.96
1.9	1.00	0.00	−22.54	74.20	−31.19	83.11
2	1.00	0.00	−26.31	78.06	−35.02	87.06

**Table 11 children-09-00807-t011:** Academic performance dimensions after matching.

		Fifth Grade				Eighth Grade			
		Mean	SD	t	*p*	Mean	SD	t	*p*
Total score	Only child	540.5	123.8			493.7	118.9		
	Non-only child	494.8	129.2	9.383	0.000	468.1	111.6	6.375	0.000
Literacy and writing	Only child	532.8	143.4			537.2	136.1		
	Non-only child	481.0	150.1	9.168	0.000	510.5	132.2	5.679	0.000
Reading and accumulation	Only child	538.0	123.9			505.3	112.5		
	Non-only child	492.8	128.1	9.305	0.000	483.8	106.4	5.607	0.000
Assignment	Only child	557.3	112.1			461.3	137.2		
	Non-only child	518.2	121.2	8.673	0.000	429.7	128.3	6.828	0.000

**Table 12 children-09-00807-t012:** Teacher-student relationships with intimacy, conflict, and supportive dimensions after matching.

		Fifth Grade				Eighth Grade			
		Mean	SD	t	*p*	Mean	SD	t	*p*
Intimacy	Only child	5.74	1.95			5.04	1.94		
	Non-only child	5.66	1.92	0.978	0.328	4.90	1.87	2.064	0.039
Conflicting	Only child	6.60	2.66			5.02	2.49		
	Non-only child	6.53	2.58	0.772	0.440	4.97	2.48	0.560	0.576
Supportive	Only child	6.48	2.19			5.02	1.95		
	Non-only child	6.35	2.14	1.480	0.139	4.91	1.90	1.628	0.104
Satisfaction	Only child	5.65	1.95			5.00	1.75		
	Non-only child	5.48	1.93	2.220	0.027	4.97	1.75	0.636	0.525
Teacher–student relationship	Only child	6.21	2.09			5.03	1.93		
	Non-only child	6.09	2.02	1.522	0.128	4.96	1.89	1.040	0.298

**Table 13 children-09-00807-t013:** Emotional management dimensions after matching.

		Fifth Grade				Eighth Grade			
		Mean	SD	t	*p*	Mean	SD	t	*p*
Within the individual	Only child	5.40	2.41			5.19	2.63		
	Non-only child	5.03	2.42	3.940	0.000	4.63	2.56	6.165	0.000
People	Only child	5.34	1.97			5.40	1.81		
	Non-only child	5.03	2.00	4.093	0.000	4.99	1.71	6.630	0.000
Adaptability	Only child	4.99	1.27			5.29	1.45		
	Non-only child	4.80	1.27	3.976	0.000	5.01	1.39	5.648	0.000
Stress management	Only child	5.40	2.19			5.43	2.02		
	Non-only child	5.06	2.21	4.072	0.000	4.92	1.94	7.443	0.000
General level of emotional management	Only child	5.34	1.97			5.41	1.90		
	Non-only child	5.01	1.98	4.311	0.000	4.96	1.81	6.947	0.000

**Table 14 children-09-00807-t014:** Peer relationship dimensions after matching.

		Fifth Grade				Eighth Grade			
		Mean	SD	t	*p*	Mean	SD	t	*p*
Peer anxiety	Only child	5.19	1.38			5.00	1.37		
	Non-only child	4.96	1.36	4.326	0.000	4.87	1.33	2.799	0.005
Peer acceptance	Only child	5.44	2.52			5.16	2.39		
	Non-only child	5.13	2.44	3.260	0.001	4.92	2.37	2.847	0.004
Peer relationship	Only child	5.29	2.02			5.10	1.98		
	Non-only child	5.03	1.97	3.325	0.001	4.91	1.93	2.655	0.008

## Data Availability

The data presented in this study are available in Appendix A.

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
