# Peer review of "Differences in Cognitive and Non-Cognitive Results between Only-Child and Non-Only-Child Children: Analysis of Propensity Scores Based on Large-Scale Assessment"

_children, 2022, doi:10.3390/children9060807_

Round 1

Reviewer 1 Report

The title of the article is: Differences in Cognitive/Non-Cognitive Results Between Only-Child and Non-Only-Child Children: Analysis of Propensity Scores Based on Large-Scale Assessment. This is a 16 page article. The study focuses on the language and emotional/relational abilities of 5-grade and 8-grade, only-child and non-only-child children. Below are all the comments and requests for clarification regarding your study.

Reviewer 2 Report

Introduction:This is appropriate and can be an interesting topic that provides helpful insight about the potential of mobile application and health habits of adolescents, but some revisions are needed to be clearer and to improve the quality of the article.

Overall I think that the literature review can focus on more update reference and research and concrete data about the topic of the article and not begging with a historical background (27 to 43 lines), that is not focus on sample and main participants of this article that are only child or not only child Chinese. Also it is important of a more deep and careful review of the literature in this topic and taking into account different cultural and social backgrounds. It is important to be more clear the if in China, studies of only children have focused on a variety of outcomes (which ones) or not.Regarding cognitive outcomes specify what are you reeling evaluating? How you measure the cognitive outcomes? Other variables should be considered.

Discussion and conclusion:

I felt that I need more information to highlight the importance that you give to the cognitive outcomes and how this can be influenced by other variables. Also more update references that corroborate your findings.

Since you study didn´t demonstrate results after the intervention implemented with healthy

How do you explain some results taking into account the confounding covariates that can contribute to a less robust propensity- matching models.

Organization of the paper, grammar, and references: Another read through to check grammar errors and format would be helpful. There were several grammar/spelling errors.

Round 2

Reviewer 1 Report

The authors responded to all the comments and reformulated the article according to them

Reviewer 2 Report

The comments and suggestions have been taken into account in improving the quality of the article, and the authors present their reply to each of them. The revisions have been very helpful in quality presentation and content in the different parts of the article.